**Data Availability Statement:** All relevant data are within the manuscript and its Supporting Information files.

**Funding:** The author(s) received no specific funding for this work.

# Transversal sero-epidemiological study of *Bordetella pertussis* in Tehran, Iran

**Gaelle Noel** [1], **Farzad Badmasti** [2], **Vajihe S. Nikbin** [2], **Seyed M. Zahraei** [3], **Yoann Madec** [4], **David Tavel** [4], **Mohand Aït-Ahmed** [5], **Nicole Guiso** [1], **Fereshteh Shahcheraghi** [2], **Fabien Taieb** [1,4]*

**1** Institut Pasteur, Center for Translational Research, Paris, France, **2** Department of Bacteriology, Pertussis Reference Laboratory, Pasteur Institute of Iran, Tehran, Iran, **3** Center for Communicable Diseases Control, Ministry of Health and Medical Education, Tehran, Iran, **4** Institut Pasteur, Emerging Diseases Epidemiology Unit, Paris, France, **5** Institut Pasteur, Centre for Translational Science, Clinical Coordination, Paris, France

☯ These authors contributed equally to this work.
‡ FS and FT also contributed equally to this work.
* fabien.taieb@pasteur.fr

## Abstract

### Objectives

Pertussis remains endemic despite high vaccine coverage in infants and toddlers. Pertussis vaccines confer protection but immunity wanes overtime and boosters are needed in a lifetime. Iran, eligible for the Expanded Program on Immunization that includes the primary immunization, implemented two additional booster doses using a whole-cell vaccine (wPV) at 18 months-old and about 6 years-old. Duration of protection induced by the wPVs currently in use and their impact as pre-school booster are not well documented. This study aimed at assessing vaccination compliance and at estimating the duration of protection conferred by vaccination with wPV in children aged < 15 years in Tehran, Iran.

### Methods

Detailed information on vaccination history and capillary blood samples were obtained from 1047 children aged 3–15 years who completed the 3 doses-primary pertussis immunization, in Tehran. Anti-pertussis toxin IgG levels were quantified by ELISA.

### Results

Compliance was very high with 93.3% of children who received the three primary and 1st booster doses in a timely manner. Timeliness of the 2nd booster was lower (63.3%). Rate of seropositive samples continuously and significantly increased from 1–2 to 5–6 years after 1st booster attaining 30.4% of children exhibiting serological sign of recent contact with *B. pertussis*. Second booster dating back 1 or 2 years was associated with high antibody titers, which significantly decreased within 3 years from injection. Among children who received 2nd booster injection more than 2 years before serum analysis, seroprevalence of pertussis infection was 8.4% and seropositivity rate was higher from the 10 years-old group.

**Competing interests:** The authors have declared that no competing interests exist.

## Conclusion

Seropositivity in children aged 6–7 years with no 2nd booster supports the need for a vaccination at that age. Adolescent booster may also be considered.

## Introduction

Pertussis, also known as whooping cough, is a highly contagious disease that remains a major cause of morbidity and mortality in infants worldwide despite vaccination [1]. The etiological agent, *Bordetella pertussis*, is a Gram-negative human-restricted bacterial pathogen that is transmitted through droplets and causes a prolonged respiratory disease that is frequently fatal in young infants.

In an effort to fight the disease, whole-cell pertussis vaccines (wPVs), made of chemically or heat-inactivated bacterium, were developed and vaccination was introduced in the 1950s and largely distributed to low- and middle-income countries (LMICs) by the Expanded Program on Immunization (EPI) since the 1970-80s. Generalized vaccination has greatly reduced pertussis burden [1, 2] but has also modified pertussis epidemiology. Adolescents and adults became new reservoirs for a disease formerly believed to be pediatric, threatening newborns too young to have completed their primary immunization [1, 3]. This was mainly due to waning immunity after vaccination with wPV, as well as after natural infection and several studies reported a duration of protection induced by primary vaccination and one booster of 5 to 10 years [4–6]. Today most high-income countries have replaced wPVs with less reactogenic acellular pertussis vaccines (aPVs) for primary immunization and booster doses, while many LMICs still use the less expensive wPVs provided through the EPI, often limiting their vaccination program to primary immunization in infants [1, 7].

wPVs may suffer from disparities in production processes [1], and studies performed on wPVs available in 1980s and 1990s showed large heterogeneity associated to highly variable immunogenicity and efficacy [8]. Importantly, information is scarce about more recently produced wPVs currently used in LMICs.

*B. pertussis* is still endemic and most recent estimates indicate that around 24 million cases and 160 000 deaths from pertussis still occurred in children younger than 5 years of age in 2014 around the world, despite high, although <90%, vaccine coverage [2, 9]. However, surveillance of the disease is rarely implemented in LMIC and these numbers are probably underestimated [7].

Iranian pertussis immunization schedule, that partially relies on EPI, includes 3 primary doses (at 2, 4 and 6 months of age) and two booster injections at 18 months and in pre-school children using a trivalent diphtheria tetanus whole-cell pertussis combined vaccine (DTwP) [10]. Pertussis vaccines locally manufactured by Razi Vaccine and Serum Research Institute were replaced by vaccines licensed by the Serum Institute of India (SII) in 2007, and since 2014, the trivalent diphtheria tetanus wP combined vaccine (DTwP) is restricted for boosters while the pentavalent diphtheria tetanus wP *Haemophilus influenzae* hepatitis B combined vaccine (DTwP-Hib-HepB) is given for primary immunization. Immunogenicity and duration of protection induced by these specific vaccines is unknown [11–13]. In addition to quality of the vaccine, completion (i.e. number of doses) and compliance (i.e. age at injection) with recommended immunization schedule impact vaccine effectiveness. In Iran, information about compliance with Iranian immunization schedule is poor, vaccine coverage is reported to be high with 1st and 3rd dose coverage over 95% since 1995, which is above global coverage estimated

to be around 85% for a 3-dose course of a pertussis-containing vaccine in 2017 [14]. Despite high coverage and delivery of booster doses, circulation of *B. pertussis* has been documented in Iran, mainly in children but also in adolescents and adults [15, 16]. Sero-surveillance data are scarce, and laboratory diagnosis aren't always obtained using the recommended technology. Accurate epidemiological data are needed to monitor and adapt current vaccination strategies.

Evaluating serological anti-pertussis toxin (PT) immunoglobulin-G (IgG) levels is a straightforward and cost-effective way of estimating occurrence of infection by *B. pertussis* [17–20]. This study aimed at applying this method to evaluate circulation of *B. pertussis* and duration of protection induced by wPV in children 3–15 years of age in Tehran, and at assessing compliance with national recommendations.

## Methods

### Study population and design

The study was implemented in 15 centers in Tehran, Iran (9 kindergartens, 4 primary schools, 1 medium school and 1 medium/high school), between December 2016 and February 2017. In order to recruit in centers representative of Tehran schools, a random selection was performed from the list of schools provided by the Ministry of Education in 6 municipality districts also randomly selected among the 22 existing in this province. Enrollment criteria were: being aged 3 to 15 years, having completed pertussis primary immunization (3 first injections) and owning detailed information about history of pertussis immunization. Three year old was chosen as lower limit of age for enrollment in the study to include children being at least a year away from 1st pertussis booster dose. Eight age groups were defined (3&4, 5&6, 7&8, 9, 10, 11, 12&13 and 14&15 years old). Date of birth, gender, pertussis immunization history obtained from record booklet from the child or from a copy from school (information required at school entry), and serology results were collected, filled in paper questionnaires and recorded in a computerized database. No information on recent history of pertussis or respiratory illness was collected.

### Pertussis immunization schedule analysis

Iranian pertussis vaccination schedule includes a 3 doses-primary immunization at 2, 4, and 6 months of age, and 2 booster doses. Recommended age for 1st booster dose is 18 months, recommended age for 2nd booster dose has varied over time and was 4–6 years before 2009, 6 years in 2009–2013 and is 5–6 years since 2014. Compliance with vaccination schedule was analyzed based on the national recommendations at time of vaccination, exact definition used was the following: having received the 1st dose at 2 months +/- 14 days, the 2nd and 3rd doses at 4 to 10 weeks interval from the previous dose, the 1st booster dose at 18 months– 2 / + 8 weeks and the 2nd booster dose at age [4–6], 6, or [5, 6] years when given in 2008, in 2009–2013, or in 2014–2016, respectively [21].

### Blood sample collection

One capillary blood sample (200–400 μl) was collected for each participant by a nurse using lancet needle (BD Sentry 23G, Becton Dickinson). Tubes were inverted several times, stored at 15–25°C and sent within 6 hours to the bacteriology laboratory at the Institut Pasteur of Iran in Tehran. At the laboratory, blood was immediately spun, and serum was stored at -20°C for later analysis. Quantitative assessment of anti-PT IgG was performed using a purified PT-containing Enzyme-linked immunosorbent assay (ELISA) kit (EUROIMMUN; reference EI 2050-G) [22], and the World Health Organization (WHO) reference serum available from the

National Institute for Biological Standards and Control (NIBSC). Results were expressed in International Units (IU)/ml. Lower limit of quantitation (LLOQ) was defined as 5 IU/ml.

### Serology analysis

Anti-PT IgG levels rapidly wane overtime and cut-off level of 40 IU/mL was used to identify seropositive individuals either following an immunization or, in absence of recent immunization, having had a contact with circulating bacteria sometime in the past year. Anti-PT IgG levels $\geq$100 IU/mL were also represented and, in the absence of recent vaccination, were interpreted as a sign of infection sometime in the past 6 months [17, 23, 24]. Duration of protection was evaluated by comparing proportions of positive individuals between i) age groups and ii) groups defined based on timespan since last vaccination among children who received either 4 or 5 injections.

### Statistical analysis

For description of continuous variables, the median, interquartile range (IQR), minimum and maximum were estimated. For categorical variables, percentages were estimated, and comparisons were conducted using the two-sided $chi^2$ or fisher exact test depending on the sample size. Logistic regression models were used to identify factors associated with anti-PT IgG $\geq$ 40 IU/ml. Statistical analyses were carried out using Stata 15 software. Differences were considered significant at $p < 0.05$.

### Ethical consideration and safety of participants

The study protocol was reviewed and approved by the Research Ethics Committee of Pasteur Institute of Iran. Informed written consent was obtained from all parents/legal guardians, and oral consent form was obtained from children aged over 7. Only participants who agreed to participate were included in the study. Authorization for data processing has been obtained from French legal authority (*Commission Nationale Informatique et Liberté* [CNIL]), and pseudonymization of names was performed assigning a code specific to the study to each participant.

## Results

### Study population description

A total of 1047 children were enrolled, and the number of children ranged from 96 to 176 in the different age groups (S1 Table). Enrolment took place over a 3-month period, and started in kindergartens enrolling youngest individuals and ended in high schools with the oldest children. In addition, boys and girls tended to be enrolled from distinct sites due to unisex schools. Thus, age had a strong collinearity with site, month of inclusion and gender.

All children received the 1st pertussis booster dose scheduled at 18 months of age. The 2nd pertussis booster had been received by 56/142 (39.4%) and 174/176 (98.9%) children belonging to the 5&6 and 7&8 age group, respectively, and all older children.

### Compliance with Iranian pertussis immunization schedule

**Compliance with recommended age or interval.** Overall median age at each of the injections matched with Iranian pertussis immunization schedule (Table 1).

Compliance with recommended age or intervals were analyzed for each dose (Fig 1). Regarding the primary immunization, no child received the 1st injection too early and only 15/1047 (1.4%) received it too late as compared to Iranian standards. Nearly all of those who had

**Table 1. Age description at pertussis vaccine injections.**

| | Primary immunization doses | | | Boosters | |
|---|---|---|---|---|---|
| | 1st (m) | 2nd (m) | 3rd (m) | 1st (m) | 2nd (y) |
| Recommended age | 2.00 | 4.00 | 6.00 | 18.00 | [4–6] if <2009 |
| | | | | | 6 if 2009–2013 |
| | | | | | [5–6] if >2013 |
| median age | 2.00 | 4.00 | 6.03 | 18.00 | 6.14 |
| 25%–75% | 1.97–2.03 | 3.97–4.07 | 5.97–6.07 | 17.93–18.10 | 5.92–6.58 |
| 10%–90% | 1.93–2.09 | 3.90–4.13 | 5.90–6.16 | 17.80–18.23 | 5.75–7.23 |
| min—max | 1.57–38.20 | 3.02–40.07 | 4.92–41.93 | 8.13–54.23 | 5.09–7.52 |
| N | 1047 | 1047 | 1047 | 1047 | 815 |

Values are presented in months (m) or years (y).

received the 1st dose on time also received the 2nd and 3rd dose on time (1004/1047; 95.9%) and were therefore compliant with full course of primary immunization. Among those who were not timely vaccinated sometime in their primary immunization course (n = 43), only a minority of individuals received a 2nd or 3rd dose too early (1/43) or too late (27/43). All children with a delayed 1st or 2nd dose received the subsequent doses within correct intervals as national standards.

Regarding compliance with 1st booster, 1000/1047 (95.5%) children have received the 1st booster at the recommended age. Of note, among those who were late at 1st booster dose (n = 21), 16 (76.2%) respected a 12 months interval from the last primary immunization dose.

A total of 517/817 (63.3%) children received the second booster dose at the recommended age. Eligible children where those already vaccinated with that dose (n = 815) and unvaccinated children aged ≥ 7 (n = 2).

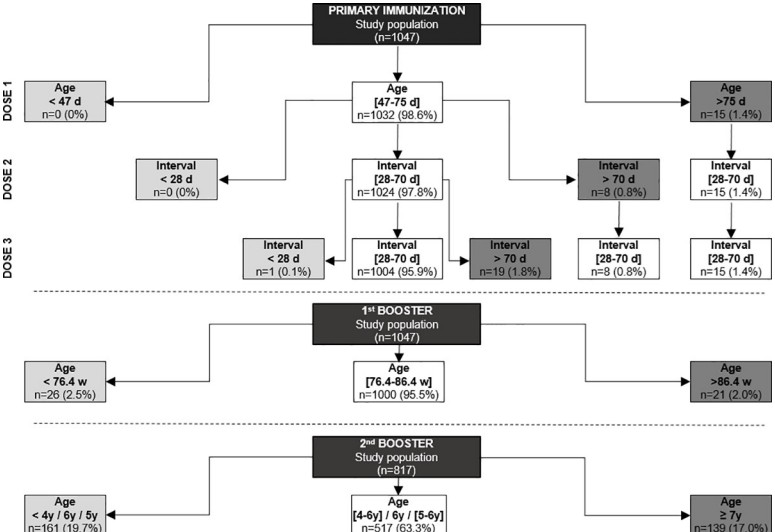

**Fig 1. Compliance with Iranian pertussis immunization schedule.** Numbers (n) and proportions (%) of children who received early (light grey), on time (white) or late (dark grey) their pertussis vaccine dose as compared to Iranian recommendations are shown for each injection. Respect to intervals in between 1st and 2nd, and 2nd and 3rd are shown with respect to compliance with 1st and 2nd dose, respectively. Days (d); weeks (w); years (y).

**Compliance with entire immunization schedule.** Overall, 702/1047 (67.1%) children were vaccinated in a timely manner (including 4 or 5 injections based on the child age), and 977/1047 (93.3%) children received the 3 primary doses and 1st booster in a timely manner.

## Serological analysis

**Description of anti-PT IgG levels.** Anti-PT IgG titers were determined for 1010 serum samples. Other 37/1047 samples were not tested due to low volume or hemolysis. Proportion of children with high IgG titers was significantly higher in 5&6 and 7&8 age groups, corresponding to recommended age for receiving 2nd booster dose.

**Anti PT IgG level in individuals who did not received the 2nd booster.** Among the study population tested for anti-PT IgG, 226/1010 (22.4%) children were not vaccinated with 2nd booster. Their anti-PT IgG levels were evaluated with respect to timespan since 1st booster dose. Shortest timespan since 1st booster injection was 17.9 months. Level of antibodies was associated to and increased with the delay since 1st booster injection categorized in year (Fig 2 and Table 2). Overall, time since 1st booster was significantly associated with higher anti-PT IgG titers (global $p = 0,007$). Rate of seropositive children among those who received the 1st booster dose 5 to 6 years before blood testing was significantly higher as compared to those who received their booster within 1 to 2 years, 7/23 (30.4%) and 1/42 (2.4%), respectively (Table 2; $p = 0.009$). These individuals were 6.5–7.0 years old. Antibody levels were not associated to compliance of these individuals with the primary and 1st booster dose schedule (Table 2).

**Anti-PT IgG levels related to 2nd booster.** A total of 784/1010 (77.6%) children had already received the 2nd booster dose. The proportion of individuals with anti-PT IgG $\geq$ 40 IU/ml was high among those who received the 2nd booster within a year and within 1–2 years

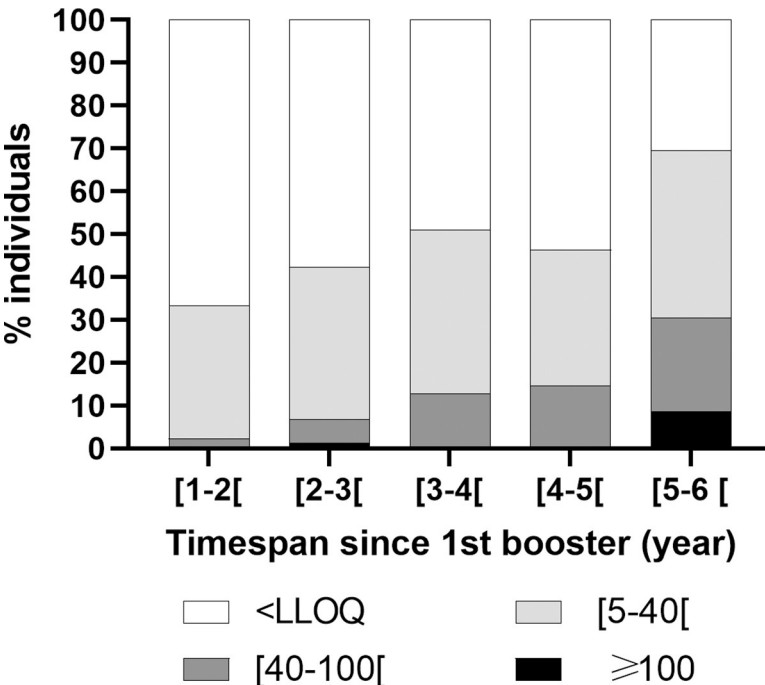

**Fig 2. Distribution of anti-PT IgG levels by timespan since 1st booster among individuals who received 4 injections.** Timespans were categorized by year.

**Table 2. Distribution of positive individuals by timespan since 1ˢᵗ booster dose among individuals who received 4 injections.**

|  |  | N | anti-PT IgG ≥ 40 IU/ml n (%) | OR (95% CI) | p | global p |
|---|---|---|---|---|---|---|
| Timespan from 1st booster (years) | [1–2[ | 42 | 1 (2.4) | 1 | - | **0.007** |
|  | [2–3[ | 73 | 5 (6.9) | 3.0 (0.3–26.7) | 0.322 |  |
|  | [3–4[ | 47 | 6 (12.8) | 6.0 (0.7–52.1) | 0.104 |  |
|  | [4–5[ | 41 | 6 (14.6) | 7.0 (0.8–61.2) | 0.077 |  |
|  | [5–6 [ | 23 | **7 (30.4)** | **17.9 (2.0–157.7)** | **0.009** |  |
| Compliance (Iranian standards) | no | 10 | 1 (10.0) | 1 | - | na |
|  | yes | 216 | 24 (11.1) | 1.1 (0.1–9.3) | 0.913 |  |
|  | Total | 226 | 25 (11.1) |  |  |  |

(27.7% and 32.5%, respectively), and significantly decreased among children vaccinated since 2–3 years (11.1%) (S2 Table).

**Anti-PT IgG levels related to infection post-2ⁿᵈ booster.** To distinguish infection-from vaccine-induced anti-PT IgG, analysis of proportion of children with anti-PT IgG ≥ 40 IU/ml was performed only in children whose 2ⁿᵈ booster injection was dating back >2 years (n = 621). Proportions of children were calculated based on age groups and antibody level cut-offs (Fig 3). Infection-related seroprevalence (anti-PT IgG ≥ 40 IU/ml) was 8.4% in this population. Children aged 9 years showed the lowest proportion of anti-PT IgG (2.4%) (Table 3). Proportion of seropositive individuals was higher in 7&8 (15.3%; p = 0.004), 10 (10.0%; p = 0.023), 11 (10.5%; p = 0.018) and 14&15 (10.0%; p = 0.026) age groups. Compliance with Iranian pertussis immunization standard was not associated to serology (Table 3).

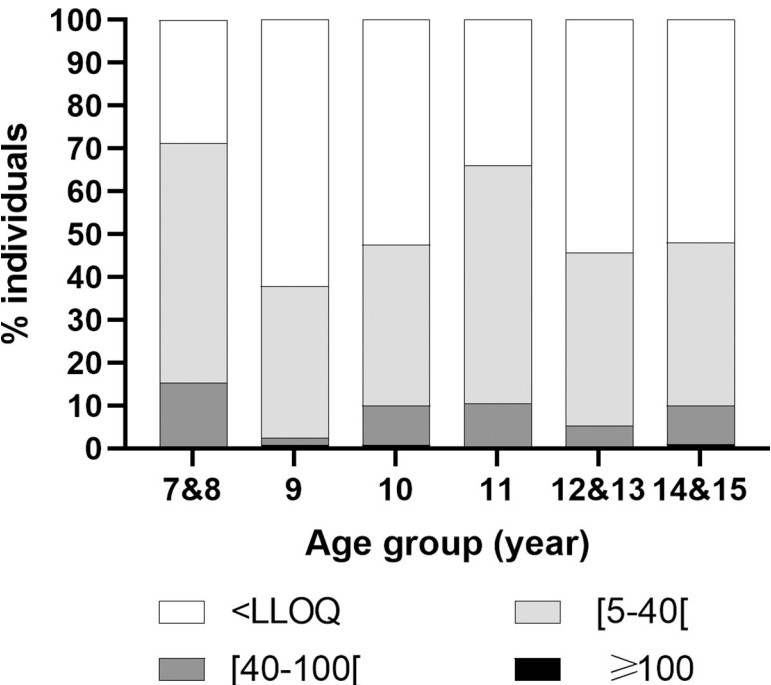

**Fig 3. Distribution of anti-PT IgG levels by age groups among fully vaccinated individuals since >2 years.**

**Table 3. Risk factors associated to positive serology among fully vaccinated individuals since > 2 years.**

| | | N | anti-PT IgG $\geq$ 40 IU/ml n (%) | OR (95% CI) | p | global p |
|---|---|---|---|---|---|---|
| **Age group (years)** | **7&8** | 59 | 9 (15.3) | **7.3 (1.9–27.9)** | **0.004** | **0.035** |
| | **9** | 124 | 3 (2.4) | 1 | - | |
| | **10** | 120 | 12 (10.0) | **4.5 (1.2–16.3)** | **0.023** | |
| | **11** | 124 | 13 (10.5) | **4.7 (1.3–17.0)** | **0.018** | |
| | **12&13** | 94 | 5 (5.3) | 2.3 (0.5–9.7) | 0.271 | |
| | **14&15** | 100 | 10 (10.0) | **4.5 (1.2–16.8)** | **0.026** | |
| **Compliance (Iranian standards)** | **no** | 313 | 23 (8.3) | 1 | - | na |
| | **yes** | 308 | 26 (8.4) | 1.0 (0.6–1.8) | 0.988 | |
| | **Total** | 621 | 52 (8.4) | | | |

## Discussion

This study aimed at evaluating age-specific distributions of anti-PT IgG in Iranian children for whom few data exist. This is the first sero-epidemiological study in children who received a pertussis vaccination that includes a booster injection, using a whole cell vaccine, at the age of 6 years.

Completion (i.e. having received all recommended doses) and compliance (i.e. having received recommended doses in a timely manner) are critical for maximizing vaccination effectiveness [1]. The delivery of 3 primary doses of wPV in children less than 12 months of age has been continuously increasing in Iran since EPI implementation in 1984 [21, 25]. The Iranian national monitoring system estimate coverage to be 95% since 1995 for dose 1 and 3, and coverage with the 1st booster dose to be 98–99% since 2014 [25, 26]. To our knowledge, no data report coverage of the 2nd booster dose yet. Our study does not give insight into proportion of children having incomplete primary immunization since receiving these 3 doses was part of our inclusion criteria. However, we observed that all participants received the 1st booster dose and 99.8% (1045/1047) of them were fully vaccinated by the age of 7.5 years. These data tend to show that once a child starts its pertussis immunization program, he/she successfully receives all five recommended doses. Our study also showed that 95.9% of the children, timely received the 3 primary doses, and compliance at 1st and 2nd booster injections was 95.5% and 63.3%, respectively. However, it should be noted that 2nd booster vaccination in Iran is performed at school entry which could explain short delays based on the month of the year the child was born.

Compliance observed herein was substantially higher that what has been reported so far in Iran. A study performed on 3610 children living in suburbs of five Iranian megalopolises, reported that 34.9%, 25.0%, 17.7% and 2.7% of children were vaccinated on time with 1st, 2nd, and 3rd injections, and 1st booster, respectively; not vaccinated children represented <5% for each dose [27]. Data from another province than Tehran indicated that about 55% of children were timely vaccinated with the injection 1, 2 or 3, with a coverage greater than 99% [28]. Compliance definition used in these studies was slightly different to the one used herein, but, when applied to our data, did not significantly impact the result. However, this study may represent highly compliant sub-population as only children having completed the primary immunization, living in Tehran, attending school and owning their vaccination record booklet were considered. Also, socio-economic and spatial disparities are known as determinants of vaccine coverage and compliance and may contribute in part to result diversity across these studies [27, 29, 30].

Duration of protection induced by the SII vaccine distributed in Iran is unknown and may differ from wPVs from other manufacturers [31]. Detection of high antibody titers at distance from vaccination has been used to indirectly identify a contact with *B. pertussis* bacteria [1, 24].

Among subjects who received primary vaccinations and one booster only, a significant increase in sero-positivity rate was evidenced at 5–6 years from 1st booster injection, evidencing waning immunity since last vaccination. The duration of protection conferred by 3 doses-primary and a 18 months-booster vaccination with Sanofi Pasteur wPV used in France in the past was defined as 9 years [6, 32]. However, data from Russia and The Gambia, using different wPV, evidenced increasing risk of infection at around 6 years old in vaccinated individuals receiving the same number of doses, such as herein, underlining the importance of 6 years old-booster in controlling infections in children aged 5 to 7, as recommended by WHO [1, 18, 20].

We observed higher anti-PT IgG titers in children who received the 2nd booster within 2 years. Anti-PT IgG thereafter rapidly decreased to reach significantly lower levels in individuals who were 3 years from their last vaccination. An important pitfall in pertussis serology is that it is impossible to differentiate vaccine induced anti-PT IgG titers to those induced by natural infection. However, our results suggest substantial immunogenicity of the vaccine and an antibody decay from vaccination over a 2 years period. Although variability exists, previous studies evaluating other wPVs used for primary and one 18 months-booster injection suggest that immune responses to PT after vaccination are negligible or reach low levels within a year [33–37]. In Iran, an additional wPV booster at 6 years of age, rarely implemented in LMIC, is performed and humoral response to a 2nd booster is unknown. The 2 years-period from vaccination to antibody decay observed herein may indicate that this second booster promotes higher persistence of anti-PT IgG. Monitoring of anti-PT IgG kinetic following 6 years old-wPV booster injection would be suitable.

Among children who received two vaccine boosters, serological sign of contact with *B. pertussis* in the past year was the lowest in the 9 years old age group and was higher thereafter ranging from 2.4% to 15.3% (Fig 3 and Table 3). These observations may indicate greater susceptibility to pertussis in adolescents. Overall number of seropositive individuals was low but demonstrated the presence of *B. pertussis* circulation among these children. During the study period, Iranian CDC reported a total of 239 and 271 suspected cases in Tehran in 2016 and 2017, respectively (unpublished data). Age information was not available, however among 347 patients for who nasopharyngeal samples were collected during the period 2004–2008 in 18 different provinces for performing pertussis diagnosis, 82% were <6 years old and only 43 were older patients for whom only 4 were confirmed positive by PCR [16]. This may reflect, at least in part, the difficulty in identifying suspected cases in older children and adults that may be caused by the milder pertussis syndrome in infected individuals of that age.

The implementation of an additional booster in adolescents may further contribute to reduce pertussis incidence in vaccinated population and lead to a shift in age of susceptible individuals. However, its impact on decreasing morbidity and mortality in infants < 1 year old, the most vulnerable population, might be limited. For this purpose, maternal immunization and the "cocooning" are two alternative strategies being recommended in countries using the wPV [1, 38]. Whatever the strategy, the availability of an aPV is required and its implementation feasibility would need to be assessed.

Our study has several limitations. First, most of the recruited children were vaccinated in a timely manner and probably do not reflect general Iranian children. However, this study shows that vaccination program is effective and that 2nd booster vaccination is appropriate, regardless of compliance. Second, no information was gathered regarding pertussis-related

symptoms that children may have experienced in the past year. This would have been interesting as occurrence of asymptomatic or mildly symptomatic infections in vaccinated individuals is greatly discussed [39]. However, questioning about history of clinical symptoms suffers from memory biases in such study design and may lead to wrong interpretations.

## Conclusions

This study shows that *B. pertussis* is circulating among children and adolescents in Tehran, despite very good compliance with the national immunization schedule observed herein. Anti-PT humoral immune response induced by the SII wPV waned over time after the 18-months booster injection and support WHO recommendations in implementing a booster in pre-school children [1]. Furthermore, an additional booster injection could be needed to reduce the circulation of *B. pertussis* in adolescents.

## Supporting information

**S1 Table. General description of the study population stratified by age category.** Values are shown as n (%).
(TIF)

**S2 Table. Distribution of anti-PT IgG levels by timespan since 2nd booster dose among fully vaccinated individuals.** Anti-PT IgG levels are shown using <LLOQ, [5–40[, [40–100 [and ≥100 IU/ml cut-offs, and the ≥40 IU/ml cut-off alone to include all anti-PT positive individuals. Anti-PT IgG titers are shown as n (%).
(TIF)

## Acknowledgments

We thank all the children and parents who kindly agreed to participate to this study, and staff from educational institutions. We also thank Ms. Sandra Corre who participated to laboratory staff training, study implementation and monitoring.

## Author Contributions

**Conceptualization:** Mohand Aït-Ahmed, Nicole Guiso, Fabien Taieb.

**Data curation:** Gaelle Noel, Farzad Badmasti, Vajihe S. Nikbin, David Tavel.

**Formal analysis:** Gaelle Noel, Farzad Badmasti, Vajihe S. Nikbin, Yoann Madec, Fabien Taieb.

**Investigation:** Farzad Badmasti, Seyed M. Zahraei, Fereshteh Shahcheraghi.

**Methodology:** Seyed M. Zahraei, Yoann Madec, Mohand Aït-Ahmed, Nicole Guiso, Fabien Taieb.

**Supervision:** Nicole Guiso, Fereshteh Shahcheraghi, Fabien Taieb.

**Validation:** Gaelle Noel, Nicole Guiso, Fabien Taieb.

**Writing – original draft:** Gaelle Noel.

**Writing – review & editing:** Gaelle Noel, Seyed M. Zahraei, Yoann Madec, Mohand Aït-Ahmed, Nicole Guiso, Fereshteh Shahcheraghi, Fabien Taieb.

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
