## [Decision Letter · Decision Letter 0]

16 Jul 2020

PONE-D-20-08084

Transversal sero-epidemiological study of Bordetella pertussis in Tehran, Iran

PLOS ONE

Dear Dr. NOEL,

Thank you very much for submitting your manuscript "Transversal sero-epidemiological study of Bordetella pertussis in Tehran, Iran" (#PONE-D-20-08084) for review by PLOS ONE. As with all papers submitted to the journal, your manuscript was fully evaluated by academic editor (myself) and by independent peer reviewers. The reviewers appreciated the attention to an important health topic, but they raised substantial concerns about the paper that must be addressed before this manuscript can be accurately assessed for meeting the PLOS ONE criteria. Therefore, if you feel these issues can be adequately addressed, we invite you to submit a revised version of the manuscript that addresses the points raised during the review process. We can’t, of course, promise publication at that time.

We look forward to receiving your revised manuscript.

Kind regards,

Abdallah M. Samy, PhD

Academic Editor

PLOS ONE

**Journal Requirements:**

**Reviewers' comments:**

Reviewer's Responses to Questions

**Comments to the Author**

1. Is the manuscript technically sound, and do the data support the conclusions?

Reviewer #1: Yes

2. Has the statistical analysis been performed appropriately and rigorously? 

Reviewer #1: Yes

3. Have the authors made all data underlying the findings in their manuscript fully available?

Reviewer #1: Yes

4. Is the manuscript presented in an intelligible fashion and written in standard English?

Reviewer #1: Yes

5. Review Comments to the Author

Reviewer #1: Please see the attached document. In short, I would suggest a simplified analysis of the data and clearer presentation of results. Although I have said yes to the appropriate use of statistics, I recommend some changes...

6. PLOS authors have the option to publish the peer review history of their article (what does this mean?). If published, this will include your full peer review and any attached files.

Reviewer #1: **Yes: **Rudzani Muloiwa

---

## [Author Response · Author response to Decision Letter 0]

7 Aug 2020

Dear Editor-in-chief,

We are grateful to the editor and reviewer for carefully reviewing our manuscript entitled “Transversal sero-epidemiological study of Bordetella pertussis in Tehran, Iran”, and for the constructive comments. We have implemented the comments and suggestions and wish to submit a revised version of the manuscript that will fulfill the PLOS ONE criteria. 

Changes in the initial version of the manuscript are either highlighted for added sentences or strikethrough for deleted sentences in the “track changes” version. Below, we also provide a response for each comment (in blue), explaining how we have addressed it. 

We look forward to hearing from you on the outcome of this second assessment. 

Best regards,

On behalf of the co-authors

Fabien Taieb, MD, MPH

• General comment

This is an important study that deals with two important and related issues on pertussis vaccination. It reports on vaccine coverage and timeliness of receiving vaccine doses, as well as adds to our knowledge of immune responses to a whole cell pertussis vaccine in a low and middle-income country setting with time from booster doses. The authors must be commended for taking on this work.

I have given specific comments that if considered I believe would improve the final manuscript. Although these seem numerous, they are all very related.

Authors’ Response:

We are grateful for the careful reading of our manuscript, and we very much appreciate your suggestions, which have been very helpful in improving the manuscript. 

• Abstract

While I appreciate the difficulty involved in writing abstracts with limits on word count, I still believe that the results section must reflect the results themselves, not just p-values in the absence of data on which the hypotheses were tested. In the presence of word limits, the authors must decide what the most important findings to reflect in the abstract are. In this case the authors still have more words to play with.

Authors’ Response:

We thank you for the advice, the abstract has been edited to make full use of the length and to better reflect with the result data.

• Introduction

Line 49. I suggest changing ‘developing world’ to low and middle income countries (LMICs) as done lower done

Line 56 ‘aPV’s’ first mention, write in full

Authors’ Response: 

Thank you for your attention, these changes have been implemented.

• Methods

Line 10. Two minor issues. Is there a rationale for these groupings? 

Authors’ Response: 

Yes, there is. A total of 100 children were recruited by age group. The classification of age groups was performed so as to finely determine the most adequate age group for a booster vaccination, thus increasing statistical power among these expected age groups. Also, three years old was chosen as lower limit of age for enrollment in the study in order to include children being at least a year away from 1st pertussis booster dose scheduled at 18 months.

Secondly, the structure of the age separation has the potential of creating an ambiguity. E.g. 3-4, 5-6 means that there is a missing ‘4-5’ in the middle. May be easily resolved with 3&4, 5&6 which closes an ‘apparent’ one year gap between the groups.

Authors’ Response: 

Thank you for the suggestion, age group writing has been modified throughout the manuscript (eg. from “3-4” to “3&4”) to eliminate any ambiguity. 

Line 135. ‘in case of contact with bacteria’ should this not better read, “in the absence of recent vaccination”…?

Line 141. I think the authors meant to say ‘continuous’ rather than ‘quantitative’?

Line 142. ‘Categorical’ rather than qualitative??

Authors’ Response: 

We are thankful for these suggestions, which have all been implemented in the Method section. 

Line 145. Although one can choose any p value cut off, by convention a p value ‘less than’ (<) 0.05 is typically chosen rather than “equal or less than”, and when chosen, it is two sided which I am assuming is what was used here for Fisher and Chi

Authors’ Response: 

The reviewer is correct, and method section related to statistical analysis was edited as suggested. We specified the use of “two-sided chi2 or fisher exact test”, and we edited the p value cut-off for considering significance from “p≤0.05” to “p<0.05”. 

• Results

158. Is there a reason for this collinearity to exist between the mentioned variables?? They don’t on the surface of it seem to be potentially collinear…

Authors’ Response: 

This collinearity is a due to enrollment strategy and school organization in Iran. “Enrolment took place over a 3-month period, and started in kindergartens enrolling youngest individuals and ended in high schools with the oldest children. In addition, boys and girls tended to be enrolled from distinct sites due to unisex schools. Thus, age had a strong collinearity with site, month of inclusion and gender”. The quoted argument has been added to the text to help the reader understands the origin of these collinearities. 

• Table 2.

§ Although I eventually figured Table 2 out, it is not an easy table and carries a significant risk of causing confusion. Unless the authors can defend it, I would suggest using a (flow) diagram instead to show this… Of course, the fact that I figured it out in the end means it can be understood, just not very intuitive. I only started getting better clarity when I drew it for myself as below….

Authors’ Response: 

We agree Table 2 is not an easy Table to understand. The proposed flowchart model indeed improves the understanding, thank you for drawing the example. Table 2 has been updated that way, making it Figure 1. Thus, Figure and Table numbers have been updated throughout the manuscript.

Lines 206-207. Is the timeliness of the second booster dose based on time interval or on expected age to receive the dose?

Authors’ Response: 

Timeliness of second booster is based on expected age. Text has been edited to remove ambiguity as the following: “A total of 517/817 (63.3%) children received the second booster dose at the recommended age.”

Lines 222, 228, etc. May I suggest consistency in the use of n (%) as the authors have been

doing throughout the document.

Authors’ Response: 

Thank you for your attention, these changes have been implemented throughout the document.

Line 227. The findings being compared should be shown and not just the p value (I am assuming this is coming from a logistic model in Table 3?). Again, I would suggest showing the data for line 230 instead of stating a p value in the absence of the compared data.

Authors’ Response:

Yes, p value that was reported line 257 referred to the logistic regression in Table 3. Text has been edited adding information on the data themselves and specifying the table to refer to, to guide the reader, as following: “Rate of seropositive children among those who received the 1st booster dose 5 to 6 years before blood testing was significantly higher as compared to those who received their booster within 1 to 2 years, 7/23 (30.4%) and 1/42 (2.4%), respectively (Table 3: p=0.009). “

Regarding line 230, and based on PLOSONE editor feedback and data sharing requirements, p value related to the association of seropositivity and compliance reported as “data not shown” was deleted. This information was implemented in Table 3, and text now refers to Table 3 in which p value can be found.

• Table 3 (now reported as Table 2 in the revised manuscript)

§ It is not clear what the p value of 0.007 means and how one interprets it (especially when next to another one of 0.104 and the OR CI that breaches the null value of 1). Is the p-value of 0.007 for the overall fit of the model or for the trend?

Authors’ Response:

We acknowledge that the p value=0.007 was not clearly defined in the table and not explained in the text. It represents the global p value, thus demonstrating that overall the time since 1st booster had an impact on anti-PT IgG titers. Table 3 has been updated to read “global p” and the value has been moved upward in order not to look associated to a specific age category. Also, the text was revised to report this data explanation, as following: “Overall, time since 1st booster was significantly associated with higher anti-PT IgG titers (global p=0,007)”. 

§ I would suggest getting rid of effect measures (odd ratios) altogether as I doubt their usefulness especially given such poor precision of estimates (wide CI). Including this only serves to undermine the observed trend of increase in proportion of cases with

higher titres.

Authors’ Response:

Although we agree that estimates have poor precision, we would prefer keeping the effect measures as they give information. However, in case you firmly believe it is better to get rid of that information, we will delete it. 

In addition, we updated effect measures values by getting rid of the 2nd decimal, thus simplifying and homogenizing with Table 4.

Lines 244/245. Already made the point about p-values and shown data. Applies to whole Section

Authors’ Response:

Text has been edited by deleting p values, and by reporting the data themselves and citing the related Table.

• Table 4. (now reported as Table 3 in the revised manuscript)

§ I am not sure what has been adjusted for in the logistic model… This is crucial because the adjustment is supposed to lead to a conclusion of age cohort being somehow independently associated with infection… It creates for a very difficult explanation (which is not available in the discussion) that in my understanding is external to the available data (such as the local epidemiology of pertussis)

Authors’ Response:

The only adjustment we were able to make was for the timeliness of vaccination as recommended by Iranian standards. However, in univariate analysis, this variable was not associated with seropositivity (p=0.988), and the adjustement in multivariate analysis did not substantially affect seropositivity results by age groups. We thought multivariate analysis data were useful, but it may indeed bring misunderstanding. Given that there is no association between timeliness and seropositivy, we decided to getting rid of multivariate analysis in Table 4 to ease the reading and to show similar format for Tables 3 and 4. In addition, to homogenize fully Tables 3 and 4, global p value and total number of children and seropositivity rate were added in Table 4. 

§ Apart from age 9 having the lowest prevalence, is there any other rationale for making this the baseline?

Authors’ Response: 

Our rationale for making age 9 the baseline was that the denominator of the first age group (7&8 years old), chosen by default, was significantly reduced (from n=165 to n=59) due to the exclusion of children whom last vaccination dated back <2 years. This led to reducing the statistical power for all the comparisons and the precision of the estimates. Then the upcoming age group (9 years old), which also included the highest number of children, was defined as baseline.

§ Is this Table necessary or as a minimum, is the search for association shown here indicated and meaningful? Would a simple description not suffice and the limitations to analyzing this noted in the discussion??

Authors’ Response:

We think that Table 4 brings additional and useful information such as seropositivity estimates, significance of association etc., which cannot be found in Figure 2, and we would prefer keeping it in the manuscript. However, the table has been revised to make it smoother (please refer to previous comment).

• Discussion

Lines 302/302. Are the authors suggesting that increased titres as a marker of recent infection is also a proxy for waned immunity? How would the titres of someone with intact immunity responding to an acute infection differ from those of someone with waned immunity?

Authors’ Response:

We understand very well this comment. We did not mean that the response of someone with intact immunity differ from those with someone with waned immunity. We edited the sentence according to this very good comment, as the following: “Detection of high antibody titers at distance from vaccination has been used to indirectly identify a contact with B. pertussis bacteria.”

---

## [Decision Letter · Decision Letter 1]

17 Aug 2020

Transversal sero-epidemiological study of Bordetella pertussis in Tehran, Iran

PONE-D-20-08084R1

Dear Dr. NOEL,

We’re pleased to inform you that your manuscript has been judged scientifically suitable for publication and will be formally accepted for publication once it meets all outstanding technical requirements.

Kind regards,

Abdallah M. Samy, PhD

Academic Editor

PLOS ONE

---

## [Editor Report · Acceptance letter]

24 Aug 2020

PONE-D-20-08084R1 

Transversal sero-epidemiological study of Bordetella pertussis in Tehran, Iran 

Dear Dr. NOEL:

I'm pleased to inform you that your manuscript has been deemed suitable for publication in PLOS ONE. Congratulations! Your manuscript is now with our production department. 

Kind regards, 

on behalf of

Dr. Abdallah M. Samy 

Academic Editor

PLOS ONE